# From *"one big clumsy mess"* to *"a fundamental part of my character."* Autistic adults' experiences of motor coordination

**Emma Gowen\*, Louis Earley, Adeeba Waheed, Ellen Poliakoff**

Division of Psychology, Communication and Human Neuroscience, School of Health Science, Faculty of Biology, Medicine and Health, The University of Manchester, Manchester Academic Health Science Centre, Manchester, United Kingdom

\* emma.gowen@manchester.ac.uk

## Abstract

Altered motor coordination is common in autistic individuals affecting a range of movements such as manual dexterity, eye-hand coordination, balance and gait. However, motor coordination is not routinely assessed leading to undiagnosed and untreated motor coordination difficulties, particularly in adults. Few studies have investigated motor coordination difficulties and their impact from the viewpoint of autistic people. Therefore, the current study used FGs and thematic analysis to document the experience of motor coordination difficulties from the viewpoint of 17 autistic adults. Four main themes were identified. First, motor coordination difficulties were pervasive and variable, being present life-long and within multiple movements and affecting many aspects of life. Furthermore, the nature of the difficulties was variable within and between participants along with differing awareness of coordination ability. Second, participants described motor coordination as an active process, requiring concentration for most actions and at a level seemingly greater than other people. Third, motor coordination difficulties impacted upon social and emotional wellbeing by placing strain on relationships, prompting bullying and exclusion, putting safety at risk and causing a range of negative emotions. Fourth, in the absence of any support, participants described multiple learning and coping strategies. Findings highlight how it is essential to address the current lack of support for motor coordination considering the significant social and emotional consequences described by our participants. Further investigation of motor learning and interactions between sensory and motor performance in autistic adults is also warranted.

## Introduction

Motor coordination involves the control of different muscles within or between limbs in a cooperative manner to achieve a movement. It is an integral part of daily life, essential for activities such as moving around, communicating, eating, playing, hobbies and working. Consequently, difficulties with motor coordination can impact negatively on various aspects such as daily living skills, quality of life, education, employment and mental health [1–4]. This study

**Data Availability Statement:** In terms of data sharing, there are ethical restrictions as the data contain sensitive information and is potentially identifiable. In addition, when reviewing the

consent forms we only have permission to share anonymous quotes in the publication and do not have permission to share the transcripts. The data availability statement has been updated to reflect this. We are aware that it is still currently unusual for Qualitative data to be made openly accessible for reasons similar to above. However, we will be giving consideration to this in future work and know of sites that publish transcripts (UK data service: https://ukdataservice.ac.uk/learning-hub/qualitative-data/). This may be easier for interviews, where you can get specific consent from each participant but it is more challenging for focus groups where some but not all participants may consent. Removing excerpts from those who did not consent can change the context/meaning of the transcripts.

**Funding:** Funding was awarded to Emma Gowen through the MRes Experimental Psychology degree programme, The University of Manchester to cover research costs for Louis Earley and Adeeba Waheed. The funders had no role in study design, data collection and analysis, decision to publish, or preparation of the manuscript.

**Competing interests:** The authors have declared that no competing interests exist.

focusses on motor coordination difficulties in autistic adults, a less explored feature of autism than the communication and social interaction difficulties that autism is typically characterised by [5]. It is estimated that at least 80 out of every 100 autistic individuals show poorer motor coordination compared to non-autistic individuals [6–8]. These motor coordination difficulties are present from infancy to adulthood [7, 9–11] and affect a range of movements including fine motor skills involving finger movements, eye-hand coordination such as reaching and grasping and gross motor skills such as balance and gait [9, 12–14]. In recent years, there has been increased acknowledgement within the research community that motor coordination difficulties are a neglected but potentially important aspect of autism [15, 16]. Studies have begun to explore whether there are unique patterns of autistic motor behaviour that can be used in diagnosis [17–19], the possible mechanisms [20–22], the relationship with social functioning and autism characteristics [23–25] and potential interventions for motor difficulties [26–28]. However, research into motor coordination in autism is still at an early stage and it remains the case that motor coordination difficulties in autistic individuals are not routinely assessed, diagnosed or treated [6, 29] and their aetiology is unclear. This situation has arisen partly because autism research has focussed on social functioning, but another reason may be because a rich description of coordination difficulties and their impact from the viewpoint autistic people is lacking leading to their importance to the lives of autistic people being overlooked. Therefore, the current study used focus groups (FGs) and thematic analysis to document the experience of motor coordination difficulties from the viewpoint of autistic adults.

Compared to children, there are relatively few studies that have examined motor coordination difficulties in autistic adults which reflects the more general situation that autistic adults are underrepresented in terms of support, diagnosis and research [30]. Quantitative studies using motor tasks coupled with sensitive measures such as motion tracking, force and button presses highlight altered movement kinematics such as increased jerkiness of arm movements [31], altered postural and gait control [12, 13], slower movement and reaction times [9, 11, 32–35], altered grip force control [36] and increased intraindividual variability of arm, finger and foot movements and gait [9, 11, 35]. Virtually nothing is known about autism in adults greater than 45 years [37], although one recent study that examined the Bruininks–Oseretsky Test of Motor Proficiency in autistic and non-autistic adults aged 40 and 60 years reported that the gap in motor coordination remains in older adulthood. As motor coordination declines with age in the general population [38] and there is suggestion of increased rates of Parkinsonian symptoms in autistic adults [39–41], it is relevant to understand how motor coordination difficulties in autistic individuals interact with ageing processes. Therefore, it was considered important that the current study included viewpoints of both younger and older autistic adults.

Although quantitative approaches provide essential and detailed insight into the how different characteristics of movement are affected, qualitative approaches (e.g., FGs or interviews) are also required to understand how autistic people experience motor coordination, what factors influence coordination ability and the impact of motor coordination difficulties. Qualitative techniques allow researchers to explore ideas to greater depth, helping to guide future quantitative studies investigating aetiology and to explore attitudes and thoughts around topics such as the potential need for support. To date, only a few studies have explored motor coordination in autistic adults using a qualitative approach (Table 1). Robledo, Donnellan and Strandt-Conroy [42] described an action theme where, participants discussed difficulties controlling, starting and stopping actions along with unstable balance, proprioceptive issues and difficulties combining two or more movements or actions. Welch et al. [43]) also identified problems with starting, stopping, coordinating and controlling movements as well as proprioception. Bloggers in their study further described frustration and anger at not being able to control their movements as intended. Participants interviewed by Bertilsson et al. [44]

**Table 1. Methodological details of qualitative studies that have examined motor coordination in autistic adults.**

| Study | Participants | Methods | Analysis |
|---|---|---|---|
| Robledo, Donnellan and Strandt-Conroy, 2012 | 5 autistic adults (19–57 years) | Interview | Constant, comparative approach where data was collected, analysed then further data was collected. Data was coded using descriptors of the sensory and motor topics (e.g. perception-vision). |
| Bertilsson et al. 2018 | 11 young autistic adults (16–22 years) | Interviews | Deductive approach using a framework based on motor control and body awareness |
| Welch et al. 2021 | 40 self-identified autistic individuals | Blog posts | Content analysis with a codebook of a priori themes |

reported a lack of awareness, understanding and control of what the body/limbs were doing, as well as muscle aches and pains. They also described difficulties in bilateral coordination between the right and left limbs and unstable balance.

Summarising this small body of qualitative work, various motor coordination difficulties are an important aspect of everyday experience for autistic individuals. However, this previous work is limited to small groups of mainly young adults or poorly characterised samples and lacks an in-depth examination of motor coordination issues that can be obtained when using more inductive approaches and thematic analysis. Consequently, a full description of the motor coordination difficulties, their impact and the use of strategies from the lived experience of autistic individuals is lacking. In the current study, FGs were conducted with autistic adults across a broad age range in order to understand how they experience and describe motor coordination across their life span, the impact of motor coordination issues on daily living skills, social and emotional wellbeing and individuals experience of learning new motor tasks and using strategies. FGs were employed as they allow opinions to be collated from a relatively large sample, compared to one-to-one interviews, and have been successfully conducted with autistic adults in previous research [15, 45–48]. Furthermore, interactions between members in a FGs allow researchers to understand the range of opinions as well as the level of agreement about topics [49], which is particularly suitable for the current study's aims.

## Materials and methods

Please see supporting information (S1 Checklist) COREQ checklist (COnsolidated criteria for REporting Qualitative research).

### Participants

Participants were recruited through the Autism@Manchester mailing list and social media outlets, University bulletins, the University Disability Advisory and Support Service and local autism support groups. Inclusion criteria consisted of (i) formal diagnosis of an Autism Spectrum Condition, checked by asking participants to provide a diagnosis letter; (ii) ability to express themselves verbally or through writing; (iii) 18 years or above; (iv) ability to attend a Zoom video or text chat FG (v) no neurological movement disorder (e.g. Parkinson's Disease or Multiple Sclerosis). It was decided not to exclude other neurodevelopmental, mental health conditions or physical health conditions (e.g. Attention Deficit Hyperactivity Disorder (ADHD), Obsessive Compulsive Disorder, depression, Musculoskeletal conditions) as these frequently co-occur with autism [39, 41], but additional diagnoses were recorded (see below). The advert highlighted that we were interested in hearing from those who had motor coordination difficulties as well as those who are unsure or did not think they had difficulties. This was to explore whether some participants become more aware of any coordination issues during the FG discussions. The study received ethical approval from The University of

Manchester's Research Ethics Committee (2021-11157-18980) and participants provided written informed consent.

Twenty-five participants were recruited of whom 8 participants withdrew before data collection for unknown reasons (5), data security concerns (1), researcher error emailing the incorrect survey (1) and mistakenly signing up for non-preferred format of FG (1). A total of 17 participants took part, aged 19 to 67 years (mean age 53.7 years), of whom 5 were female. Ten participants were white British, 2 mixed ethnicities, 1 other, and 4 preferred not to disclose ethnicity. Thirteen participants were educated to University degree or higher-level apprenticeship level, 2 had further education qualifications (A levels, BTEC level 3 or equivalent) and 2 participants reported "other" but did not specify. Eight participants were employed and 9 were unemployed. Five participants received formal support, 11 received no support and 1 participant did not specify. Eight participants had no other diagnoses, 3 had Developmental Coordination Disorder/dyspraxia, 3 dyslexia, 2 ADHD, 1 Diabetes, 1 Sjorgrens, 1 Ehlers-Danlos, 1 anxiety disorder, 1 depression and 2 preferred not to say. Participants were asked to complete the Adult Developmental Coordination Disorder checklist [50] to characterise the severity of motor difficulties in the sample. A total score equal to or higher than 56 indicates risk of Developmental Coordination Disorder, whereas a score equal to or higher than 65 indicates probable Developmental Coordination Disorder. The average score was 60.3 (32 to 109) with 6 participants below 56, 3 between 56 and 64 and 8 greater than 65. Eight participants scored >17 on the childhood section, indicating coordination difficulties since childhood.

## Study development and procedure

The research team consisted of four researchers who were two MRes Psychology students (LE, AW), their supervisor (EG) and collaborating researcher (EP). AW and LE have experience in learning difficulties and mental health and LE has Dyspraxia (DCD) bringing his own experiences of coordination difficulties to the project. EG is a researcher in the field of sensory perception and motor control in autism as well as lived experience of a neurological movement condition, Multiple Sclerosis. EP is a researcher in sensory perception and motor control in autism and Parkinson's disease. All had previous experience and training of conducting FGs and thematic analysis either at undergraduate/Masters level or during previous research studies.

All participants were emailed a link to the online questionnaires (consent form, ADC and demographics), created using Qualtrics. To facilitate inclusivity, participants were given the option of a Zoom video FG or a Zoom text chat FG, the latter was chosen to ease difficulties that autistic people often have with various aspects of verbal communication such as the timing of conversation turns, auditory processing, and attention [51]. Face-to-face FGs were not possible because of COVID restrictions at the time. Participants were sent an information sheet and a 'What to expect form' for the type of FG they chose. Details provided about the researchers included experience (e.g. lived experience of dyspraxia and professional experience of autism) and qualification that the data was being collected for (Masters). Prior to the study start, all participant facing material (adverts, information sheets and the 'what to expect' document) were reviewed by the Autism@Manchester expert by experience group.

A total of 5 FGs were conducted in June and July 2021 (Table 2), each lasting 2 hours with a short break halfway through. Ten participants took part in a Zoom video FG and 7 took part in a Zoom text chat FG. Participants were randomly allocated to a FG depending on their availability to attend and their preferred format. Based on our previous experience of conducting FGs with autistic adults [15, 45, 48] it was decided to cap FGs at 5 participants to allow all participants to regularly contribute within the 2 hour session, without feeling rushed or overwhelmed. The result of the cap and participant attrition was that some groups had less than 4

**Table 2. Demographics of the 5 FGs (FG).** Note that the FG refers to the chronological order in which they were completed. Where a participant ID is associated with a *, this indicates that the participant scored <56 on the ADC.

| FG, type and participants | N (number of females) | Mean age (range) in years | Mean ADC (range) |
|---|---|---|---|
| FG 1: Zoom Video (P4,* 12,* 13,16*) | 4(1) | 43(23–65) | 49(46–56) |
| FG 2: Zoom Text Chat (P7,8,11*) | 3(2) | 36(19–51) | 70(41–109) |
| FG 3: Zoom Video (P5,18, 19,23, 24*) | 5(1) | 42(22–67) | 61.8(32–77) |
| FG 4: Zoom Text Chat (P1,6) | 2(0) | 52(47–58) | 82(72–92) |
| FG 5: Zoom Video (P15,* 22,25) | 3(1) | 56(40–67) | 65.7(48–77) |

participants, commonly viewed as the minimum size for a focus group [52]. The FGs were led by one member of the research team (LE or AW), accompanied by EG to assist with running the sessions, asking follow up questions and monitoring participants for distress. Notes were made by all researchers during and after each FG.

The facilitator followed a semi-structured schedule (see S1 File), consisting of 8 key questions asking about: (1) The way participants define and refer to motor coordination difficulties; (2) The kinds of motor coordination difficulties they experience; (3) How coordination difficulties affect daily experience; (4) Whether coordination is affected by anything; (5) How well participants learn movement related tasks; (6) How motor coordination had changed over time; (7) Whether any participants had received support for coordination difficulties; (8) Whether participants thought motor coordination difficulties were related to their autism or other diagnoses. Questions were developed through discussion with the research team and discussion around motor coordination with members of the Autism@Manchester expert by experience group. Researchers were given time to ask follow-up questions to allow free-flowing, meaningful conversation. Participants were sent a debrief sheet after the FGs via email, as well as vouchers for participation.

## Analysis

The Zoom Video FGs were audio recorded, then transcribed by an external university approved service for intelligent verbatim transcription. Transcripts were not returned for comment to participants. Participants were pseudonymized in both video and text chat FG transcripts and the researchers had access to information that could identify individual participants during and after data collection. Three researchers (EG, LE, AW) used reflexive thematic analysis to analyse the data, taking an inductive, semantic and realist approach. Therefore, the researchers approached data without prior theoretical assumptions and without reference to unarticulated meaning or social context, allowing the experiences of participants to be understood from their perspective [53, 54]. This approach, which allowed the dataset to be richly described was thought appropriate due to the limited previous research on motor coordination difficulties in autistic adults.

The three researchers independently coded all of the data, following the six-step technique advocated by Braun and Clarke [53, 54] and meeting regularly to discuss their views and interpretations of data. Firstly, the accuracy of each transcript was checked against the original recordings. The researchers familiarized themselves with the data by re-reading the transcripts independently whilst making any initial notes of key ideas then meeting to discuss the main topics they had noted. The second phase involved re-reading and line-by-line coding of the transcripts to identify features (words, sentences or paragraphs) of the data related to the scope of the study. This was done in Microsoft Word using the comment tool. Researchers began by using semantic coding. This meant explicit, surface level meanings from data were coded to

keep codes as close to the data as possible. As researchers became familiar with data, some latent coding involving interpretation of the data was also done. Researchers met twice to refine and agree on the codes (S1 Table). In the third step, each researcher used the codes to independently construct meaning-based themes done by collating similar or related content and ideas from the data into themes using Microsoft Excel. Researchers met to discuss theme development. Two researchers had four themes whereas one researcher had six themes, but further development and discussion led to agreement of all researchers upon four final themes (S1 Fig). The fourth phase reviewed the allocated themes against the dataset as a whole to ensure that the themes captured all relevant aspects of the data. Themes were appropriately named and given a short definition in the fifth phase and the researchers met to agree names and final themes. Additionally, a detailed analysis of each theme led to the allocation of multiple subthemes in two of the themes which were deemed large and complex. The final phase involved bringing together the themes and supporting data in a report. Feedback on the themes was not requested from participants.

The sample size was based on recommendations by Braun and Clarke who indicate that data saturation is less applicable to reflexive thematic analysis when interpreting data where "new meanings are always possible" as opposed to codebook or coding reliability types of analysis which often have predefined themes and use broad topic summaries [55]. We took a pragmatic approach, choosing a feasible sample number that would allow data to be collected and analysed within the time constraints of the Master's project. In addition, the research team believed that the data was likely to be very rich as the question was highly focussed, the sample contained participants with the condition of interest (autism) and each person was likely to contribute a good amount "information power" (i.e. relevant information). Finally, our previous work on similarly focussed topics has produced rich and sufficient data with a comparable number of participants [45, 48].

## Results

Following analysis, four main themes were identified (Fig 1).

### Theme 1: Motor coordination difficulties are pervasive and variable

Motor coordination difficulties were present within multiple movement types and activities and therefore affected many aspects of life. However, the nature of the difficulties was variable within and between participants. Three subthemes describing the nature of motor coordination issues, their spiky profile and variable awareness and acceptance were identified.

**Nature of motor issues.** At the start of the FG, participants were asked about how they referred to motor coordination difficulties and how they related to the terms gross and fine that are commonly used in research. Generally, participants referred to coordination issues as "*clumsiness"* or referred to the specific issues they had.

> "I think of difficulties to do with anything that involves coordination or spatial awareness maybe. For example walking, hand-eye coordination, writing, tying shoelaces, playing a musical instrument, sense of direction"
>
> (FG2, P7)

> "I find life difficult anyway, both in terms of spatial awareness, how to move around in 3D, how to interact, etc. I am here in a physical sense, but don't feel I have fine control of the vessel I inhabit. It glitches very often by dropping things, getting confused, bumping into things, etc"
>
> (FG4, P6)

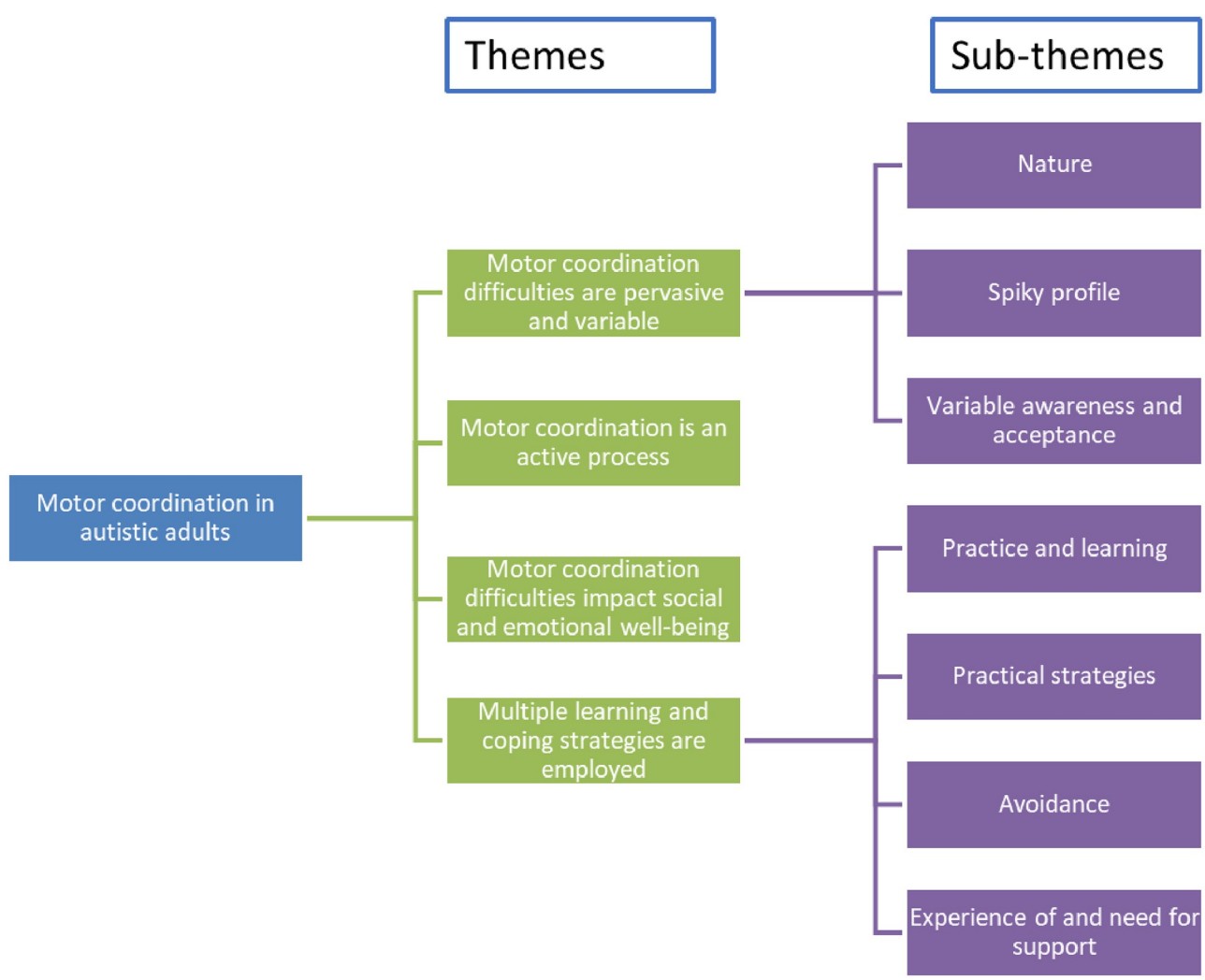

**Fig 1. Main themes (green) and sub-themes (purple) identified from the FG data.**

While some participants did not know what fine and gross movements meant, many understood gross as big movements or overall body coordination of your limbs and fine as small movements that you do with your fingers or hands. Eye-hand coordination was more difficult to define, but participants often agreed that it required a *"middle category"* (FG1, P12) and could use both fine and gross depending on the task such as when *"picking up a pencil."* (FG1, P4).

From discussion, coordination difficulties were apparent across fine and gross movements as well as eye-hand coordination. Issues with fine motor activities included sewing, doing up buttons and tying shoelaces. Gross motor problems were present with running and walking with some participants reporting that they could not *"walk in a straight line"* and tend to *"zigzag"* or *"drift"* to one side (FG 2, P7,8). There were also issues with balance, *"sometimes when walking I will sort of lose my balance and start to wobble as if I was drunk"* (FG4, P1) as well as spatial awareness including *"bumping into things, knocking things over"* (FG4, P1) and *"stumbling over things"* (FG3, P19). Consequently, participants reported experiencing trips and falls.

Regarding eye-hand coordination, participants found catching, throwing, pouring liquids and reaching difficult, leading to dropping and knocking over objects.

*". . .if I turn a page in my book while I'm working, I can very easily hit a glass, or knock something over."*

(FG1, P13)

They also expressed difficulty coordinating hands and feet, choosing which hand to use and eye-foot coordination *". . .just being able to kick straight, which is something I never actually learned to do."* (FG1, P13)

These coordination difficulties impacted a variety of activities such as getting dressed, handwriting, DIY, sports, playing instruments, carrying things, tying knots (e.g. shoe laces or bin bags) and eating (*"biting your own tongue"* FG1, P13). Actions that many people take for granted could be difficult, *"but even getting in and out of a shower I can find quite complicated"* (FG3, P5). Coordination difficulties could lead to frequent *"domestic disasters"* (FG1, P4) such as dropping meals on the floor.

Many people found driving difficult which they attributed to *"not being able to process the verbal instructions fast enough to react appropriately"* (FG2, P7), *"thinking of many things at the same time"* (FG3, P23), anxiety and *"a combination of coordination and poor reflexes"* (FG4, P1). Therefore, a combination of factors in addition to poor motor control are likely to influence driving ability in autistic individuals.

Coordination difficulties were severe enough in some participants to be noticed by other people who picked up on our participants walking styles, commented that they were clumsy or had *"an odd way of doing things"* (FG5, P15) or openly wondered whether they had dyspraxia.

*"I have had people tell me that I have a strange gait and that I am recognisable by my walk"*

(FG2, P7)

Variability in motor coordination ability was apparent between participants and not all participants shared the same difficulties and expressed relative strengths. This included being *"very good at catching things"* such as with rounders and cricket, (FG3, P5 & 18) *"being excellent at drawing and painting in fine detail"* (FG4, P1) and finding the driving experience *"wonderful. . .because it's very cause and effect, there's a flow to driving"* (FG1, P13).

There were also mixed views on how coordination difficulties had changed with time. Many thought they had improved, but this was thought to be due to using strategies:

*"I would say that my motor coordination has improved over time, mainly because I am more aware of what I can do to manage any issues"*

(FG2, P11)

Some thought that they had remained unchanged and others thought they had got worse, possibly due to ageing:

*"But I find I think I'm worse now [referring to falls], and partly it's probably because my knees are going, and that partly I think it's because I'm taking more interest in life around me, partly it's because the pavements are dreadful."*

(FG5, P12)

Nobody said that they no longer had issues, indicating that coordination difficulties in autistic individuals are likely to be present throughout life.

**Coordination can have a spiky profile.** As well as the variability between participants observed above, variability could also occur within participants. Some participants noticed that they were better (or even gifted) at particular tasks involving motor coordination while being significantly poorer at other motor tasks. Sometimes this was related to the distinction between fine and gross motor control *"I'm rubbish at gross motor skills, but better with fine"* (FG1, P4), but many expressed a spiky profile where seemingly related actions were performed to different abilities: Although, P5 was gifted at playing the piano they struggled with writing *"No, I've always had terrible writing that hardly anyone can read and well, often I can't read it."* (FG1, P5). This participant also had extreme difficulty with seemingly less complicated actions such as getting in and out of the shower and washing up which could appear like *"a conundrum, a puzzle, a contradiction"* (FG3, P19). While hand writing was also difficult for P11, they could *"do certain motor tasks, like shaving"* (FG 2, P11). A particularly striking example was around fell running and walking:

*"I have difficulty walking over uneven ground, but I fell run and run, cycle and swim."*

(FG2, P7)

Sometimes, the same activities like writing, playing sports such as basketball or snooker, can be done very well but at other times, they can be difficult. One participant expressed she can *"play snooker, sometimes very well, sometimes very badly"* (FG5, P22) and another expressed *"there's no guarantee I'll get it right the second time"* (FG5, P15) after doing a task well. P18 (FG3) even described the ability to do movements in basketball sometimes as 'fortuitous' rather than deliberate.

**Variable awareness and acceptance of motor coordination difficulties.** Some participants were very aware of their motor coordination difficulties and could clearly articulate what these were:

*"Because I kind of notice that the difference between the way I do things, in terms of, it's very stop/start, and how other people can seem to do things seamlessly."*

*(FG1, P13)*

This awareness included thoughts about possible causes such as *"proprioception, and sensory processing. . ."* (FG1, P4) or suspecting *"it's as much a learning problem as anything else"* (FG1, P16). For one participant, an injury led to him having to learn to walk again, which led to him discovering more about his coordination and improving it with help from his wife:

*"And you think right, what I was doing before was essentially carrying my weight over the balls of my feet, and moving with my torso rather than me feet."*

(FG3, 19)

For others, they thought that the way they moved was similar to other people and the motor issues had to be pointed out to them before they became aware of them:

*"But a lot of my difficulties have been pointed out to be by other people and without them doing that I wouldn't necessarily be aware of them myself—like my walk. . . . . . And I didn't believe it at first because I can't tell I'm doing it. I just assumed I was like everyone else"*

*FG2, P7*

One participant did not think they had motor coordination difficulties but then revealed that they could not "*figure out*" how to tie shoe laces until "*the first year of college*" (FG 2, P11), suggesting that they might be unaware of having motor difficulties. Participants who had less awareness tended to also have less understanding of the nature of their motor difficulties and possible causes.

> "*Very often, I talk to autistic people who have just seen somebody do something, assumed that they can do it, and then been totally puzzled as to why they fail, with no concept of the fact that this other guy may have had five years' intensive training before he did what they saw him do.*"
>
> (FG1, P16)

Participants seemed to be at different stages of acceptance with their motor coordination issues. For some, clumsiness was integrated into the self, as part of their persona or character, resulting in both negative and positive experiences:

> "*I feel like I am one big clumsy mess.*"
>
> (FG2, P8)

> "*But I've had kind of positive experiences, because I guess I've developed like a clown persona. Like, I'm always the one that falls in a bush, I've done that before, or does something, you know, clumsy and funny, but I kind of capitalise on it, and make it into a joke.*"
>
> FG1, P4

In contrast to "*leaning into it*" (FG1, P13), others tried to mask or "*shield*" (FG1, P13) from their motor coordination difficulties or felt they had acquired appropriate coping strategies to accommodate any motor coordination difficulties:

> "*As far as motor skills go, it is what it is, for me. There are issues, but they're not issues that they can be…whatever issues I've got they're kind of worked around*".
>
> FG5, P15

Many participants highlighted how having an autism diagnosis made their motor coordination easier to understand and accept, suggesting that autism diagnosis is important for accepting the motor as well as social aspects that come with autism:

> "*…prior to diagnosis[…]those kind of things could be a bit frustrating for me, but as I say, now I've got that awareness it's not really an issue*"
>
> FG5, P15

## Theme 2: Motor coordination is an active process

A consistent theme across all FGs was that motor coordination is an active process, requiring concentration for most actions and at a level seemingly greater than other people.

> "*But things other people are doing as without really trying, I realise I'm doing a lot of processing to achieve.*"
>
> (FG5, P25)

*"I think I'm very careful a lot of the time. . . . . .so that I don't very often spill things or whatever".*

(FG4, P25)

Many participants expressed how they had to actively concentrate and monitor a range of movements such as eating, sports, walking, picking objects up and even sitting in a non-optional way:

*"But all through my life, for example, sitting has always been an effort. So I'm sitting up now, and if I concentrate on doing anything at all, or if I'm trying to actually do anything I end up slouching."*

(FG5, p25)

When they were able to concentrate, activities might be successful, but when concentration was broken accidents occurred.

*"But the amount of times that I've, like, ended up with, like eating, and biting my own tongue constantly, if I'm not constantly thinking".*

(FG1, P13)

*"it's a matter of concentration in that I find I'm actually quite good at doing things that are really fiddly and intricate, because my full focus is on it. But at the same time, I can really be clumsy just walking. So, I think it's more. . .my focus is more on one thing, so I'm able to con-centrate and therefore I'm not so clumsy".*

(FG3, P23)

Some participants discussed and agreed that autistic people *"trip more often, but they're bet-ter at actually correcting it. . .you'll fall less often"* (FG3, P5). Participant 19 (FG3) said this is possibly because *"you're having to concentrate quite hard on the process"*, suggesting that the increased focus might sometimes reduce the chances of falling.

Unsurprisingly, factors that compete for concentration, made motor coordination more difficult. While general factors affecting concentration such as such as stress and performance anxiety could sometimes negatively impact actions, participants discussed more specific task related and sensory factors. This included tasks that require multi-tasking such as thinking "*of two things at the same time*" (FG1, P13) or performing coordinated movements using more than one limb at once.

Sensory issues also impacted on concentration and consequently on actions. Altered pro-prioception was described by some participants, potentially driving the need for closer moni-toring of movements or concentration on one action taking away from the sense of body in space:

*"I was concentrating on picking this thing up on the floor. And so what was happening with my other hand, completely oblivious"*

(FG3, P19)

Participants also discussed how sensory overload due to visually cluttered, busy or noisy environments disrupted ongoing motor actions.

*"For me, it's about the overall amount of the sensory input. And that could be, it could be social, as well, but just the amount that I'm having to process. And it seems like my coordination is one of the first thing to go out the window, when my brain is too full."*

(FG1, P4)

For some, the sensory and motor issues were closely entwined as *"everything's mixed up"* (FG3, P18) and *"difficult to untangle the different effects of the different sensory issues"* (FG 5, P25).

Participants also indicated that "*the time to react and think about the actual movement, has like a significant impact on how easy that movement is*" (FG1. P13) so that dynamic situations where they needed to make reactive movements were difficult and stressful:

"If *something is very dynamic and fluctuating, that's, again, a point where the clumsiness starts to set in*"

(FG1, P13).

Consequently, they often needed to plan movements to have more time to predict outcomes and often needed to "*try and do things slower and over a longer period of time in steps*" (FG4, P6).

This advance planning enabled them to have the time to fully concentrate on the actions, whereas reactive more dynamic situations provided insufficient time for the concentration required to plan and predict. Relatedly, there was a sense that although concentration could overcome some motor coordination difficulties, it was not possible to "hyper-concentrate" (FG5, P15) on some tasks and concentration could not make up for the ease of movement experienced by other people.

*"Because I was concentrating and I knew the rules of the game, I could be good at it to a certain level. But there was that natural coordination and I guess forward-thinking that I was missing".*

(FG3, P23)

As a consequence of the required concentration, many participants found motor coordination effortful and fatiguing, requiring "*a lot of mental [not physical] energy*" (FG3, P5) limiting the length of time they could perform tasks and requiring energy management strategies (see later subtheme on practical strategies).

*"Everything you do is more deliberate, so is more exhausting because of the extra concentration involved"*

(FG4, P1)

## Theme 3: Motor coordination difficulties impact social and emotional wellbeing

*"I don't think you can entirely untangle the social awkwardness from the physical awkwardness, because they are too intertwined if you get what I mean"*

(FG4, P1)

For many of our participants, coordination difficulties hindered social relationships and negatively impacted emotions. In regards to social interactions, motor coordination difficulties strained existing relationships and day to day interactions by being annoying, inconvenient or dangerous for other people:

> *"And I was in a friend's holiday house and they had a nice white carpet, and I stood there pouring Bovril over it while I was busy talking to somebody. And that didn't do much for the relationship there either."*

(FG3, P19)

> *"...if I go on my own [shopping], I can run the trolley into people by accident..."*

(FG 3, P5)

Participants expressed how others could often have negative reactions or judgments to their clumsiness, such as getting frustrated and angry, shouting at them for being slow or, accusing them of being drunk, all of which prevent the formation of or strain existing relationships. This lack of understanding by others also extended to understanding the need for help, which was particularly difficult to convey to others due to being an adult:

> *"...it's very difficult to say to people that you need help with these things, because it's so basic and for heaven's sake you're a grownup and you should be able to do it."*

(FG5, P22)

Yes, there were some examples where close friends and family were accepting and supportive and have learnt how to *"understand and accommodate it"* (FG 3, P23).

In some cases, participants were *"bullied"*, *"ridiculed"* or *"criticised"* about their *"inelegance, or impractical movements,"* walking or sports ability. Motor coordination difficulties also led to some participants being excluded from social activities such as walking trips, Physical Education at school, sports or particular sports positions (*"I was always put in defence"* FG1, P13). This was often instigated by others, but participants also highlighted self-exclusion. They excluded themselves from certain activities due to disliking the activity and finding it challenging as well as due to the emotional impact of bullying. Therefore, some social activities were *"somewhat barred off"* (FG1, P13) for our participants.

Motor coordination difficulties resulted in a large and varied impact on emotional wellbeing, partly due to the motor difficulties themselves and partly due to negative reactions of others. In terms of the direct impact, participants discussed feelings of disappointment, envy of others' superior motor skills and frustration at being unable to perform certain actions or being clumsy.

> *"And eventually you just wear yourself out with it, and there's just so much disappointment and, yeah, you're just not getting any better."*

(FG5, P25)

Coordination difficulties also put participants own safety at risk leading to anxiety and fear of conducting everyday tasks, social interactions and walking around. One participant was *"really terrified of walking around"* for fear of falling (FG 5, P22) and another *"chopped off the end"* of their finger when cooking and frequently drops or smashes crockery (FG2, P7).

The behaviour of others also impacted on emotions as highlighted by one participant describing how the negative reactions of others created embarrassment and fear of social interactions:

*"This [motor coordination issues] combined with socialising issues, I would constantly make social gaffes that left me embarrassed and other people angry, and eventually I avoided going out at all because I was afraid of the confrontations/ social awkwardness, so became a recluse."*

(FG4, P1)

The combination of the direct impact of motor coordination difficulties and the indirect responses of others resulted in poor *"self-esteem"*, a sense of self-loathing because of not being able to do some activities and feeling *"socially outcast"* due to the negative behaviour and bullying by others.

*"You can start to hate yourself. I actually shut myself away from the world for ten years because of this [motor coordination difficulties] and other issues to do with autism, ten years I'll never get back"*

(FG4, P1)

Participants raised whether some of these social and emotional experiences were influenced by gender, in that men may be more judged than women by their ability to move well in certain activities such as sport.

Although negative emotions were more commonly ascribed to motor coordination difficulties, some participants described a sense of pride and achievement after having learnt and practiced movements after a long time. Participant 7 (FG2) said *"doing things in spite of"* difficulties can be *"helpful for self-esteem"*.

## Theme 4: Multiple learning and coping strategies are employed

Participants described a range of strategies that they use to cope with motor coordination difficulties in the absence of any direct, structured support. Four subthemes of practice and learning, practical strategies, avoidance and experience and need for support were identified.

**Subtheme: Practice and learning.** Many participants talked about the need to rehearse, practice or "study" actions and of "*getting stuck in*" (FG1, P16).

*". . .I mentioned earlier, that I use, I very much, like, practice doing very mundane and trivial things, quite a lot. And that would be a kind of, how do I do this to make it look normal"*

(FG1, P13)

This can be implemented by practicing routine-based skills like picking things up or "*walking through a house and opening doors and closing them*" (P13, FG1). Similarly, another described difficulty in stepping over stiles in the countryside so practiced the movement by stepping "*up and down off this chair*" regularly in the home (FG5, P25).

Alongside practicing, participants discussed the need to pre-plan or familiarise themselves with upcoming actions. Sometimes this was through "*visualising*" or "*imagining*" doing the activity, suggesting that some people might use motor imagery. For others, this was too difficult as visualisation and performing movements requires "*two separate areas of my mind and I*

*cannot combine them"*, so performing a task before visualisation is necessary to understand it (FG4, P6). For some, discussing pre-planning with the group made them realise that they did not plan actions and perhaps *"that's what you should do to avoid these things [accidents] happening"* (FG1, P4).

There were several different approaches to learning actions. Some participants needed to be shown and actually perform the movements, while others needed to be talked through the movements and corrected as well as writing the instructions down fully. Others preferred watching videos to having lessons. Some participants described how they would break down any movement task into distinct stages so that they could process it step by step rather than simultaneously. Some participants described how others commented that they had an odd style of performing actions and would try to (unsuccessfully) teach them the "correct" style.

> *"And I think generally it's been a case of, well, don't expect me to do it the proper way, and let me do it my way and then I might stand a chance"*
>
> (FG3, P25)

Many participants indicated that with practice they could learn new motor skills and improve on existing ones, providing examples of actions such as typing, sports, musical instruments and tying shoelaces. However, this might take a long time:

> *"It took me nearly 10 years to learn to ride a bike. I am having to play catch up my whole life."*
>
> (FG4, P6)

Despite practice, participants generally thought that they could not achieve the ability of their non-autistic peers. One participant described how despite football and rugby sessions "*three times a week"* and *"being interested in learning"* he noticed that *"compared to someone who was not autistic, I still would never kind of hit the same level of coordination"* (FG 1, P13). Other participants discussed reaching a "plateau" or that they could never *"refine"* the movement, so other people would reach a higher skill level them. It was also sometimes difficult for participants to differentiate improvement due to changes to fundamental motor control mechanisms from having developed better strategies (see next subtheme), with many noticing improvement but thinking that *"the underlying skill level"* was not *"normal"* (FG1, P16)

> *". . .in one sense I haven't got more control. I've found better ways of living with myself in the world that I've got"*
>
> (FG3, P19)

To compensate for a lack of ability to improve their physical movements, some participants would learn the "*technique and tricks of the trade"* and read around the strategy of the sport they were playing in order to enhance their performance.

While many actions for participants did not become easy or natural, some participants provided examples where actions had become easy, automatic and of a good level following practice:

> *". . . the only exception would be like musical instruments, where if I do an action over and over again then it becomes quite natural, I don't have to think about it so much"*
>
> FG3, P5

One participant wondered whether autistic people might sometimes be "*better than every-body else*" on certain motor tasks because they "*have to put so much more effort in*" and "*prac-ticed so much more*" (FG3, P18).

**Subtheme: Practical strategies.** Participants described many practical strategies that helped them to feel more control over their coordination and "*more able to live*" in their world (FG3, 19). Ultimately, these strategies do not improve motor coordination skills–they simply reduce the chance of clumsiness, falls, or accidents. These included putting into place certain preventative and facilitative measures such as organising the environment around them so it is neat and uncluttered, pouring liquids over a sink, getting a handrail for the shower, using a tray instead of carrying cups, wearing cardigans with zips rather than jumpers and buttons, wearing clothes that do not show stains and use of objects as visual markers to help "*navigate*" through the environment.

Due to the amount of effort some actions required, participants used energy planning strat-egies, carefully planning the number of activities they could manage each day, such as taking a shower every other day rather than daily. They also found that reducing expectations, being patient with oneself and moving slower was helpful. Participants discussed how physical train-ing, keeping fit and healthy and specific strength exercises helped with moving around. Finally, other people could be incredibly enabling by providing help to carry things, navigate around, prevent accidents and facilitate cooking and eating.

**Subtheme: Avoidance.** In contrast to practice and "*getting stuck in*" (FG1, P16), many participants described how they avoided certain activities that were challenging and "*tended to move towards things where coordination is not really important*" and do not require "*precision*" (FG1, P12). Avoidance was due to a mixture of physical and mental effort and fear of failure and ridicule. It restricted the range of activities participants could do (e.g. sports) and as recog-nised by the participants themselves, avoidance leads to worsening of motor skills or failure to learn them:

> "*I think, truthfully, in the sense that, I think we've all admitted to some extent, shying away from activities that we struggle with, that realistically, that causes a regression in skill*".

> (FG1, P13)

**Subtheme: Experience and need for support.** All our participants had received no spe-cific support for coordination issues. They generally thought that support and "*a more detailed assessment of people's needs*" (FG 3, P5) would be valuable if it "*could encourage and build those skills up*" (FG1, P13) that they were avoiding and facilitate opportunities "*to do something that's active but not impossible for us to do*" (FG5, P22). They also thought that recognition and support "*could help people to understand why they have certain difficulties with some things*" (FG3, P5) and provide more awareness of who could help provide this support. As with diag-nosis of autism, diagnosis of coordination issues can help with acceptance and support from others:

> "*And I'm also prepared to tell people I'm dyspraxic, and I think that takes away some of the stress*"

> (FG5, P22)

They also highlighted that if they had been diagnosed earlier, they may have received sup-port earlier to develop strategies and improve motor skills. Nevertheless, participants

emphasised that support should be needs led rather than diagnosis led with individuals provided with the *"minimum amount of support"* (FG3, P5) required in order to retain a sense of independence, self-confidence or because *"clumsiness is a kind of fundamental part of my character"* (FG1, P4).

## Discussion

This FG study has for the first time provided an in-depth view of motor coordination from the view point of autistic adults, including those greater than 45 years of age. Findings reveal a wide range of difficulties affecting fine motor, eye-hand coordination, balance and moving around which subsequently impact on many basic activities of daily living. Participants indicated that motor coordination difficulties were life-long, and that improvement was more likely due to the use of strategies than a fundamental change in motor control. A particularly revealing finding was descriptions of motor coordination as effortful, fatiguing and requiring full concentration, aspects that were also noted in interviews with adults with DCD [56]. Complementing previous objective, quantitative findings [9, 38], experience of motor coordination varied between and within participants. Participants also provided a rich description of the multiple ways that poor motor coordination affected their social and emotional well-being, as well as the strategies they had learnt to adopt in the absence of formal support. In the following sections we discuss some key concepts that have been identified across the main themes.

### Altered motor learning

A number of features raised by our participants suggests difficulties with learning movements and motor-based tasks. Participants described how performing movements is an active process, requiring full concentration even for apparently simple actions, suggesting that actions have not become automatic. Indeed, when asked about their ability to learn actions, most participants indicated that learning could take a long time and that they often did not reach the same level as their peers. This contrasts with the spiky profile that some participants expressed where they could be very good at particular actions that apparently became automatic. As some examples of these spiky profiles consisted of being both poor and good at apparently similar actions, it is possible that spiky profiles reflect the benefit of many hours of practice at an enjoyed activity coupled with poor generalisation to similar movements. Alternatively, spiky profiles may be due to particular motor system being more or less affected. For example, some of our participants noted that they were relatively better or worse at gross versus fine manual motor skills which are known to use different motor circuitry [57]. Similarly, Bertilson et al. [44] reported that their participants had better fine rather than gross motor skills. This highlights the importance of future research to quantify each individual's motor coordination across different motor tasks, which together with subjective reports can be used to aid support decisions but also to understand aetiology.

A further aspect that fits with altered motor learning is increased difficulty performing more complex actions such as those involving simultaneous steps or the use of more than one limb at the same time, a finding also reported by Robledo et al. [42]. In these situations, having pre-existing motor programs that require little concentration can free up capacity for multi-tasking and monitoring the overall goal of the action or task. Similarly, automatic actions can help in fast, reactive situations when a response needs to be chosen quickly, and was a further aspect that our participants described that they struggled with. Reaction times have been consistently shown to be slower in autistic compared to non-autistic individuals [9, 11, 32–35]. A recent study using aiming movements highlights that longer reaction times are potentially due to more time required for cognitive planning mechanisms, compared to motor processes such

as time to general muscle activity [35], fitting with the idea that motor actions have not been learnt to an automatic level in autistic individuals.

Previous studies provide inconsistent evidence for disrupted motor learning in autistic individuals [58–64]. Following a review of sensorimotor adaptation and motor sequence learning tasks, Bo et al. [64] concluded that mixed results may be due to factors such as intact or enhanced motor learning for tasks that rely more on proprioceptive than visual input (e.g. 61,63), the use of explicit, rule based strategies by autistic participants and individual differences. Indeed, it seems likely that although similar levels of motor learning may be achievable for some tasks, *how* autistic individuals are learning is different [58, 60, 61, 63]. We would also add that it is clear from our participants that motor learning is possible, but is slower and more laborious and that some individuals may lack awareness that it is required (although this requires further exploration). The previous lab tasks used may have been too simple, structured and repetitive to reliably detect group differences and future motor learning tasks should employ real world tasks that are more challenging and introduce greater uncertainty about the learning environment [e.g. 65]. Our participants raised practice and learning as strategies for improving motor coordination, so understanding what factors facilitate and impede motor learning for autistic individuals is key to developing appropriate motor therapies.

## Relationship with sensory processing

Sensory symptoms such as hypersensitivity and sensory overload are estimated to occur in ~90% of autistic individuals [66]. For many of our participants, these sensory issues were closely entwined with movement difficulties so that it was difficult to separate the two. They also indicated that when they experienced sensory overload it became more difficult to concentrate and perform movement. This relationship is unsurprising considering that effective motor control relies on converting sensory signals such as vision, touch and proprioception into a motor plan and monitoring and predicting sensory feedback [67, 68]. Indeed, positive correlations between motor ability and sensory symptoms [69] or perceptual processing such as visual motion [70] have been reported. Furthermore, autistic individuals consistently experience difficulty performing motor tasks that require greater use of sensory information (e.g. pointing to targets, catching, tapping in time to an auditory beep) compared to tasks that are less reliant on sensory feedback (e.g. autonomous finger tapping and rapid hand turning) [32, 71, 72]. Linking back to the previous section on motor learning, it is possible that altered sensory input makes it more challenging to plan and predict actions leading to difficulties learning new movements [see also 9]. However, with effort and practice movements can be learnt. It would have been informative to collect questionnaire measures of sensory experiences from our participants to further understand the links between sensory and motor characteristics and is a direction for future quantitative studies that can include more detailed and objective sensory and motor data.

Some individuals also talked about their sense of proprioception being poor and this was also apparent in some of the quotations relating to a lack of awareness of their posture or limbs, unless they were specifically concentrating. Proprioceptive difficulties were reported by Robledo et al. [42] and the lack of awareness, understanding and control of body and limbs recounted by participants during interviews [44] could also reflect problems with proprioception. Studies of people without proprioception have demonstrated the importance of proprioception for accurate and controlled movements and that greater concentration is required to use other senses to compensate such as vision [73–75]. Therefore, altered proprioception in autistic individuals could cause the need for increased concentration on actions and subsequently more clumsiness of limbs where concentration is not focussed. Alternatively, the

increased need to concentrate due to reduced motor learning could reduce proprioceptive awareness of body parts not directly involved in the task. Proprioception has been relatively neglected in autism with only five studies directly measuring proprioception in autistic children and none in adults. Four of these studies found altered proprioception in autistic compared to non-autistic children [76–79], while one found no differences [80]. These results, together with our qualitative findings indicate that further investigation of proprioception is warranted in both children and adults.

## Relationship with social interactions

It was clear that motor coordination difficulties impact on social relationships for our participants. Motor and social aspects were sometimes described as inseparable in that they contributed to the persona of an individual in either a positive or negative way. Previous work in children has shown that motor ability and social skills are linked so that autistic children with poorer motor skills score more highly for autistic characteristics [81–84; for a review see 25, 85] and infants with motor difficulties are more likely to be diagnosed with autism [23]. Previous authors have suggested that motor issues may contribute to social communication difficulties through reduced participation in play and exploration of social contexts and subsequently reduced observational learning opportunities as well as delayed looking at social cues leading to missing opportunities to engage socially and interpret behaviours [23, 25, 82].

Importantly, our work with adults has provided a greater insight into the multiple ways that motor coordination difficulties can impact upon social opportunities and well-being. First, our findings highlight that the increased effort and concentration needed to perform motor movements likely takes away attention from social aspects. This could be during social interactions, but also having to decline social engagements due to fatigue built up from the effort to control movement. Similarly, the autistic bloggers in the article by Welch et al. [43] described how burnout could occur from the extreme and continual effort to initiate and inhibit movements. Second, clumsiness can be perceived as annoying or dangerous by others leading to exclusion of the autistic individual from the activity. Third, motor coordination issues can lead to self-exclusion from certain social activities due to finding the activity challenging, fear of safety when moving around or fear of negative reactions from others. As noted by our participants, this exclusion or avoidance creates a negative cycle where they experience less opportunities to practice and socialise leading to static or worsening motor and social skills and negative thoughts. Importantly, other people could be both enabling and disabling and for some participants, family and friends were accepting and supportive of their coordination difficulties. Knowledge of these multiple links between motor coordination and social interaction is informative when developing specific outcome measures for motor therapies.

## The need for assessment and support for motor coordination difficulties

While the majority of our group described experiencing motor coordination difficulties with 11 out of the 17 participants scoring at risk of or probable DCD on the ADC checklist, only 3 were diagnosed with DCD or dyspraxia reflecting the underdiagnosis reported by other studies [86]. Repetitive or stereotyped behaviours are key diagnostic criteria for autism, but this assessment does not include the range of motor coordination difficulties apparent in the current study and it remains the case that motor coordination difficulties are not routinely assessed or supported [29]. Indeed, none of our participants had received specific support for their motor coordination difficulties. Yet, as described by our participants, motor coordination difficulties resulted in problems with daily functioning, personal safety, fatigue, emotional distress and

interference with social relationships emphasising that motor assessment for adults should become a consistent element of the wider autism diagnosis and assessment procedures, as argued by others for children [29, 81, 87]. This was echoed by the comments of our participants who thought that a detailed assessment of a person's needs would be valuable along with encouragement and support to build up skilled tasks that were not impossible to do. Without available support, our participants described a number of practical strategies that they employed together with their experience of practicing and learning new movement skills. These insights could form the basis for developing informative material for autistic individuals.

It was also apparent that participants had variable awareness of their motor coordination difficulties and some required them to be pointed out by others, suggesting that many autistic people might not be aware that they have motor issues. Therefore, assessments should include both objective and subjective elements to fully understand the interplay between how a person views their movement and the reality of their functioning. Recognition of motor coordination difficulties is important as an individual may not initially make the connection between motor issues and their impact upon social and emotional well-being. As highlighted by our participants, improved awareness of motor coordination difficulties can facilitate greater understanding and acceptance of both their autism and movement. Subsequently, individuals may be more able to investigate and use strategies to overcome particular difficulties and accept that they may need to spend more time learning and practicing certain tasks. Increased awareness of difficulties may also make it easier to ask for help and for other people to understand why a person may be clumsy. Of course, during the assessment it may become apparent that a person has good motor skills or that they already have the required support. Indeed, as highlighted by our participants, any resulting support should be needs driven.

A further reason for motor assessments is that while it is clear from our participants that motor coordination difficulties are life-long, there is limited understanding of how these are affected by normal ageing processes. Previous work has reported that autistic groups have poorer balance than non-autistic groups [12] and our participants mentioned this, together with falls. As falls are the leading cause of injury, injury related disability and death in older people [88] it will be important to assess whether autistic people are at an increased risk of falls and whether this should be assessed. Interestingly, a greater than 3-fold higher rate of hip fractures in autistic children and adults has been reported [89] which could be due to falls. If falls risk is indeed higher in autistic people, this could be addressed with appropriate strength and balance exercise therapy [90]. Finally, as noted by our participants, activities of daily living are likely to require assessment and support across all ages but particularly in older adults [91].

### Limitations

While these findings provide the first in depth study of motor coordination from the viewpoint of autistic adults, they are specific to our participant sample. Although we used both written and verbal FGs to increase inclusion, our participants were relatively well educated and able to use language to express their experiences. Obtaining experiences from a wider demographic as well as those with learning disabilities would extend the generalisability of these findings. As there were a large range of Adult Developmental Coordination Disorder checklist scores, it could be suggested that those with higher scores contributed more to the FGs. However, those with a score of less than 56 on the checklist were represented across the themes, suggesting they also had motor coordination issues and similar experiences. We included participants with additional diagnoses to provide a more representative and ecological participant sample. Nonetheless, it is possible that motor coordination difficulties that they described relate more

to these other diagnoses than to motor coordination difficulties specific to autism. This would be interesting to explore in future research by separating out those with or without additional motor relevant diagnoses and comparing experiences using interviews, as well as objective measures of their motor coordination. However, it is likely that most of the descriptions of motor coordination in our participants relate to long-term difficulties as half scored >17 on the childhood section of the ADC and for those participants who had additional diagnoses, only one scored <17 on the childhood section.

## Conclusions

Motor coordination difficulties in autistic adults are wide ranging, affecting a number of different actions relating to manual and eye-hand coordination, balance and gait and impacting upon daily living skills. There is variability between participants and spiky profiles within participants. While autistic adults indicated that they can learn new motor skills, this tends to take longer and be at a lower ability level then non-autistic peers. Indeed, performing even simple movements is described as effortful. Motor coordination difficulties have a significant life-long impact on autistic people's well-being, but recognition of and support for them is currently lacking. It is essential to address this gap considering the significant social and emotional consequences of motor coordination issues described by our participants. Future avenues to explore include understanding how autistic individuals learn movements, the relationship between proprioception and motor ability in autistic individuals and the development of routine motor assessments and support for autistic adults.

## Supporting information

**S1 Checklist. COREQ (COnsolidated criteria for REporting Qualitative research) checklist detailing page numbers of recommended items that should be included in qualitative research.**
(PDF)

**S1 File. Outline of procedure and questions asked during the FGs.**
(DOCX)

**S1 Table. Details of the different semantic codes that were used to build the themes.**
(DOCX)

**S1 Fig. Theme development.**
(DOCX)

## Acknowledgments

We would like to acknowledge all the participants for their time and insightful comments given to the study. We would also like to acknowledge Eve Edmonds for checking the accuracy of each transcript against the original recordings and Leneh Buckle for providing feedback on a draft of the manuscript.

## Author Contributions

**Conceptualization:** Emma Gowen, Ellen Poliakoff.

**Data curation:** Emma Gowen.

**Formal analysis:** Emma Gowen, Louis Earley, Adeeba Waheed.

**Funding acquisition:** Emma Gowen.

**Investigation:** Emma Gowen, Louis Earley, Adeeba Waheed.

**Methodology:** Emma Gowen, Louis Earley, Adeeba Waheed, Ellen Poliakoff.

**Project administration:** Emma Gowen.

**Supervision:** Emma Gowen.

**Writing – original draft:** Emma Gowen.

**Writing – review & editing:** Emma Gowen, Louis Earley, Adeeba Waheed, Ellen Poliakoff.

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
