## [Decision Letter · Decision Letter 0]

28 Mar 2023

PONE-D-22-30953From “one big clumsy mess” to “a fundamental part of my character.” Autistic adults’ experiences of motor coordination.PLOS ONE

Dear Dr. Gowen,

Thank you for submitting your manuscript to PLOS ONE. After careful consideration, we feel that it has merit but does not fully meet PLOS ONE’s publication criteria as it currently stands. Therefore, we invite you to submit a revised version of the manuscript that addresses the points raised during the review process.

We look forward to receiving your revised manuscript.

Kind regards,

Filippo Manti

Academic Editor

PLOS ONE

Journal Requirements:

Funding was awarded to Emma Gowen through the MRes Experimental Psychology degree programme, The University of Manchester to cover research costs for Louis Earley and Adeeba Waheed. 

Reviewers' comments:

Reviewer's Responses to Questions

**Comments to the Author**

1. Is the manuscript technically sound, and do the data support the conclusions?

Reviewer #1: Partly

Reviewer #2: Partly

2. Has the statistical analysis been performed appropriately and rigorously? 

Reviewer #1: N/A

Reviewer #2: N/A

3. Have the authors made all data underlying the findings in their manuscript fully available?

Reviewer #1: No

Reviewer #2: Yes

4. Is the manuscript presented in an intelligible fashion and written in standard English?

Reviewer #1: Yes

Reviewer #2: Yes

5. Review Comments to the Author

Reviewer #1: In this study, focus group interviews were used to analyze the performance, existing impact, and corresponding support status of motor coordination disorders in autistic adults. In my view, current study will bring highlights of the problem of motor coordination in autistic adults, and will further pave the way of further understanding of motor coordination related behaviors in autistic adults. Even that, several points the author still need to consider.

1. The inclusion criteria of the subjects and the diagnosis of current study should be stated clearly. The current subjects were a sample who could express themselves in words and communicate through the media successfully. To be noticed that that sample is only the minor group of autistic. It will be great to provide the detailed diagnosis information of each subject, and the comorbid (e.g., ADHD, anxiety, depression).

2. As the author stated that “A total score equal to or higher than 56 indicates risk of Developmental Coordination Disorder, whereas a score equal to or higher than 65 indicates probable Developmental Coordination Disorder. The average score was 60.3 (32 to 109) with 6 participants below 56, 3 between 56 and 64 and 8 greater than 65.”, there were 6 participants below 56 (minimal was 32) how could the make a conclusion of autistic could experience motor coordination problem? The author should list each subject’s score in a table with what typical motor coordination problem of each include subject having. Otherwise, it will be quite confused how the subjects with low score also show motor coordination problems.

3. How did the authors get four themes? The authors should clearly state the concrete process with a flowchart.

4. Whether the number of the participants in the result part can be recoded according to the group order, such as FG1, P1-P5, or whether the specific situation of the participants can be tabulated so that the readers can quickly grasp the specific situation of the specific subjects.

Reviewer #2: Regarding case selection more than half of them had associated problems like ADHD, anxiety ,depression, Sjogren's syndrome and Ehlers Danlos syndrome. These may have additional affects on motor movements. There was no mention if the movement problems were present from early childhood or developed later or progressively deteriorating. This would have helped to understand the nature of the altered motor coordination. Further the participants status of vision and hearing and sensory profile were not mentioned in the methodology. If the author had taken the above this would have given greater clarity of the association of altered motor coordination with autism. There is no data provided of the semantic coding done to comprehend the themes.

6. PLOS authors have the option to publish the peer review history of their article (what does this mean?). If published, this will include your full peer review and any attached files.

Reviewer #1: **Yes: **Xiaoling PENG

Reviewer #2: **Yes: **Shabina Ahmed

---

## [Author Response · Author response to Decision Letter 0]

15 May 2023

Please see attached document "response to reviewers"

---

## [Editor Report · Decision Letter 1]

23 May 2023

From “one big clumsy mess” to “a fundamental part of my character.” Autistic adults’ experiences of motor coordination.

PONE-D-22-30953R1

Dear Dr. Gowen,

We’re pleased to inform you that your manuscript has been judged scientifically suitable for publication and will be formally accepted for publication once it meets all outstanding technical requirements.

Kind regards,

Filippo Manti

Academic Editor

PLOS ONE

---

## [Editor Report · Acceptance letter]

25 May 2023

PONE-D-22-30953R1 

From *“one big clumsy mess” to “a fundamental part of my character.”* Autistic adults’ experiences of motor coordination. 

Dear Dr. Gowen:

I'm pleased to inform you that your manuscript has been deemed suitable for publication in PLOS ONE. Congratulations! Your manuscript is now with our production department. 

Kind regards, 

on behalf of

Dr. Filippo Manti 

Academic Editor

PLOS ONE